# What is Where by Looking: Weakly-Supervised Open-World Phrase-Grounding without Text Inputs

**Tal Shaharabany    Yoad Tewel    Lior Wolf**
Tel-Aviv University
{shaharabany,yoadtewel,wolf}@mail.tau.ac.il

## Abstract

Given an input image, and nothing else, our method returns the bounding boxes of objects in the image and phrases that describe the objects. This is achieved within an open world paradigm, in which the objects in the input image may not have been encountered during the training of the localization mechanism. Moreover, training takes place in a weakly supervised setting, where no bounding boxes are provided. To achieve this, our method combines two pre-trained networks: the CLIP image-to-text matching score and the BLIP image captioning tool. Training takes place on COCO images and their captions and is based on CLIP. Then, during inference, BLIP is used to generate a hypothesis regarding various regions of the current image. Our work generalizes weakly supervised segmentation and phrase grounding and is shown empirically to outperform the state of the art in both domains. It also shows very convincing results in the novel task of weakly-supervised open-world purely visual phrase-grounding presented in our work. For example, on the datasets used for benchmarking phrase-grounding, our method results in a very modest degradation in comparison to methods that employ human captions as an additional input. Our code is available at https://github.com/talshaharabany/what-is-where-by-looking and a live demo can be found at https://replicate.com/talshaharabany/what-is-where-by-looking.

## 1   Introduction

> What does it mean, to see? The plain man's answer (and Aristotle's too) would be, to know what is where by looking. In other words, vision is the process of discovering from images what is present in the world, and where it is.
>
> Marr [50]

We address the task of detecting the objects that exist in a given image, without limiting ourselves to a predefined list of objects, of the type that detection algorithms employ. This property is sometimes referred to as open-world vision. Our method assigns a text phrase to each detected object. The similar task that employs an additional text input describing the image is called phrase-grounding. We describe our approach as "purely visual" to denote that there is no text input. Since the task is performed without the use of bounding boxes in any phase of training, the level of supervision employed corresponds to weakly supervised. As far as we can ascertain, the task of weakly-supervised open-world purely visual phrase-grounding is entirely novel. For short, we refer to this task, which is illustrated in Fig 1, as "what is where by looking" (WWbL).

In order to tackle this task we rely on two recent pretrained vision-language models: CLIP [60] and BLIP [41]. These models are trained on large corpora of matching images and captions and can produce a matching score between an input image and an input text. Trained on millions of such pairs, CLIP has been extensively used in zero-shot recognition [94], image generation [21, 25, 45], image

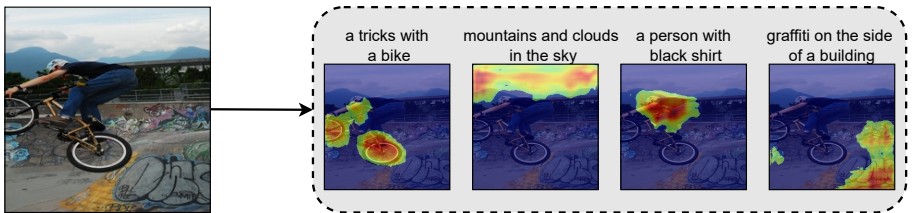

**Figure 1:** Given the image on the left, our method extracts multiple sentences that describe objects in the image, and generates a foreground mask that localizes the object

**Table 1:** Comparison to different lines of previous work on image localization.

| Name | References | Open-world | Weakly supervised | Purely visual |
|------|-----------|------------|-------------------|---------------|
| Object detection | [61] | ✗ | ✗ | ✓ |
| Phrase-Grounding | [42, 85] | ✓ | ✗ | ✗ |
| Weakly supervised localization | [66] | ✗ | ✓ | ✓ |
| Weakly supervised Phrase-Grounding | [2] | ✓ | ✓ | ✗ |
| What is where by looking (WWbL) | This work | ✓ | ✓ | ✓ |

editing and manipulation [57, 11, 24, 95], image and video captioning [53, 72, 73] and other tasks [52]. BLIP, a more recent addition, also incorporates into its training phase the ability to generate image captions. It also relies on a bootstrapped captioning model to generate matching sentences and on a bootstrapped filtering model to eliminate false matches from its training set.

While BLIP can produce a matching score, we choose to use CLIP for producing such a score and BLIP for generating plausible captions. This modular choice allows us to study the matching-based components (such as generalizing weakly supervised localization methods) separately from the WWbL. It also emphasizes the modular nature of our method. Given time and resources, we would be able to evaluate the method using only BLIP or replacing BLIP with CLIP-based image captioning methods, which are considerably slower than BLIP [73].

Our WWbL method comprises two phases. During training, an encoder-decoder $g$ is applied to extract a foreground mask for an input image, given an image caption. This is the same level of supervision that is required to train CLIP or BLIP, albeit performed on a much smaller dataset. Since no segmentation mask (or bounding box) is used, this is considered weakly supervised.

Four loss terms are used. The first one requires that the CLIP matching score of the foreground region and the matching text be maximal. The second helps ensure that the background region is unrelated to the text. The third term considers the explainability map of CLIP, given the input image and the caption [13] as a guide for the foreground mask. Lastly, in order to encourage compact foreground masks, a regularization term is added.

A two-stage inference procedure employs the learned foreground segmentation network $g$ and a pretrained-BLIP model. The first stage extracts object proposals using a Selective Search algorithm [74]. The second stage uses BLIP to generate captions for each proposal and find text clusters in CLIP's embedding space, in order to avoid repetitive descriptions. Network $g$ is then used to find the foreground mask for each sentence, given the entire image as input and the sentence.

Our results show that network $g$, as a weakly supervised localization network, outperforms the state-of-the-art. The same model also outperforms the state-of-the-art in phrase grounding. When the entire method is applied, with the two stages described above, the results obtained for WWbL are on par with the results obtained using human captions for the task of phrase grounding. These results are considerably better than those obtained by the baselines.

Our main contributions are: (i) A new state-of-the-art solution for weakly-supervised localization tasks with text and image as input, and (ii) Tackling, for the first time, as far as we can ascertain, the fundamental WWbL task, which is an open-world detection task that does not rely on any localization information during training.

## 2 Related Work

Open-world computer vision is an emerging paradigm that stands in contrast to recognition systems that can only address the limited number of classes and types of examples that were observed during training. Specifically, open-set classification [7, 86, 38] requires the ability to understand when a new sample is not from the observed classes.

Since large visual-language models are trained on hundreds of millions of samples [60], many classes of objects are observed during training. It is impractical to enumerate all object classes. Moreover, these models can recognize phrases that were unseen during training, e.g., "a green dove". Therefore, the use of the term "open-world" for such models is not based on learned vs. unseen classes, but rather on testing the method without limiting the type of objects observed during the inference.

Open-world localization methods generalize more restricted methods, which rely on a relatively short list of detectable objects. Similarly, weakly supervised methods, which employ labeling at the image level, generalize fully supervised methods that require bounding boxes during training. Finally, methods that do not receive as input guiding information regarding the content of the image (we call these "purely visual") generalize methods that receive as input a list of objects to detect or text that describes the contents of the image.

Tab. 1 compares the task we tackle to other localization tasks. As can be seen, the task we tackle -"what is where by looking" (WWbL) - generalizes all other localization tasks. Unlike supervised object detection, it is open-set and weakly supervised. Weakly supervised localization methods are not open-set. Lastly, phrase grounding methods employ a textual input, while WWbL does not.

**Object Detection**     One of the fundamental tasks of computer vision is object detection, where a closed set of object categories is localized based on training sets that contain costly bounding-box annotation [43, 20]. Since the advent of deep learning, many fully supervised algorithms have been proposed for this task [61, 26, 44]. In contrast to these methods, our approach employs image-level annotation (global text description) and an open dictionary to describe the detected object.

**Phrase Grounding**     is a weakly supervised phrase localization task, in which text phrases are associated with local image annotations and local human-annotation signal [2, 89, 33]. Many methods extract text embedding from a pretrained language model [16, 63, 35], together with image representations, to obtain a common semantic space for image-text pairs. Our model is based on CLIP [60] as a text-image relation space to extract the localization of objects. Li et al. [42] collected 27M image-text data points, of which 3M have local human annotation, and used a semi-supervised training method for phrase-grounding, which employs CLIP. It cannot be compared directly with our work, since it uses additional supervision. Arbelle et al. [3] proposed a self-supervision method scheme for generating a localization map.

**Weakly Supervised Localization (WSOL)**     Class Activation Map (CAM) explainability methods have been offered in recent years for solving WSOL tasks [92, 93, 59, 46]. Most of these algorithms train a classifier to distinguish between sub-categories of the main object (Birds, Cars, Dog etc), employing a localization loss term for the explainability map [80, 82, 51, 28, 55].

Similarly to us, Shaharabany et al. [66] employ an external localization network in addition to the classifier. During training, the classifier supplies gradients to the localization network. In contrast to our work: (1) they compare the classifier outputs with and without the foreground mask, while our algorithm compares the masked image with the text in the CLIP space, (2) there is no background loss term, and (3) explainability is not used as a signal.

**Image Captioning**     Captioning is a fundamental vision-language task. Early methods applied RNNs [49, 37]. Attention was added to identify relevant salient objects [83, 64]. Subsequently, Transformers modeled interactions among all image elements with self-attention [75, 56, 22]. Recent works have shown significant improvements in robustness and generalization by using large-scale vision-language data sets [73, 53, 41, 34, 77]. Our method employs BLIP [41], which is a vision-language pre-training framework that is trained on images and their caption, employing a method to filter out noisy data. The framework also provides an image captioning model and shows good performances in diverse image domains and for a large variety of real-world objects.

**Explainability**     Many methods generate a heatmap that indicates relevancy for CNNs, e.g., [65, 8, 47, 67]. Most relevant to our work is the use of GradCAM [65] relevancy maps as a cue for weakly

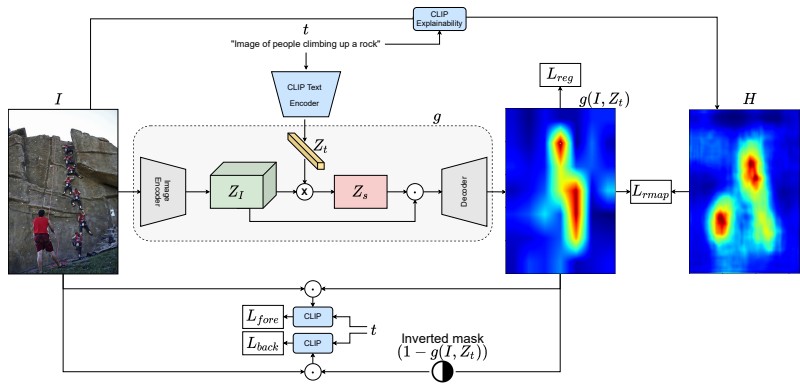

**Figure 2:** The architecture of our model and the optimization loss terms. The input image $I$ is encoded with the VGG16 encoder part of $g$, and the test description $t$ is encoded with CLIP's text encoder. The resulting text embedding multiplies the spatial dimension of the image embedding to find a similarity between regions in the image. The similarity map $Z_s$ weights the image embedding. The results are passed to the decoder part of $g$ to generate the output map. The loss term $L_{rmap}$ compares the output map to the CLIP explainability map with L2 loss. $L_{fore}$ forces the output map to focus on the text-relevant pixels by increasing the similarity between $I \odot g(I)$ and the text. $L_{back}$ aims to guide the 1-$g(I)$ to focus on the background by decreasing the cosine similarity between $I \odot 1 - g(I)$ and the text. $L_{reg}$ adds a sparsity constraint on $g(I)$

supervised segmentation methods [54, 79], including recent contributions that employ GradCAM heatmaps as pseudo labels for weakly supervised localization and segmentation algorithms[78, 28]. The literature on transformer explainability is sparse and is mostly limited to pure self-attention architectures [1, 12]. A recent work by Chefer et al. [12] fully addresses bi-modal transformer networks, and is the method employed to obtain relevancy maps. The solution is based on Layer-wise Relevance Propagation [5], with gradient integration for the self-attention and co-attention layers.

## 3 Method

In this section, we describe our method for generating a localization map based on image and text input signals without local annotation. We use our method on three different tasks (i) Weakly supervised object localization (WSOL) (ii) Weakly supervised phrase-grounding (WSG) (iii) What is where by looking (WWbL).

At the core of our method lies a segmentation network $g$, which is trained in a weakly supervised manner. The network $g$ has two inputs: an input image $I \in R^{3 \times W \times H}$, and an embedding of the input text $Z^T$. For the simpler task of WSOL, network $g$ receives only the first input $I$ as input signal. The loss terms used in both cases are the same, and, for the sake of generality we write the equations with $g$ that has two inputs.

The training of $g$ does not involve any ground truth localization information. Instead, it relies on two types of signals: (1) comparing the CLIP score of the extracted segment to that of the entire image, and (2) including the heatmap of the CLIP network, as obtained with the relevancy method of Chefer et al. [14], in the segmented region.

Four-loss terms are used, using the input image $I$, the text $t$, and the relevancy heat-map of CLIP given $I$ and $t$, which we denote as $H$. The terms are (1) a foreground loss $L_{fore}(I, t)$ (2) a background loss $L_{back}(I, t)$, (3) a relevancy heatmap loss $L_{rmap}(I, H)$, and (4) a regularization loss $L_{reg}(I, g)$.

The foreground loss encourages the segmentation network $g$ to output a map that increases the similarity between the text and the masked image $g(I, t) \odot I$, $\odot$ being the Hadamard product.

$$L_{fore}(I, t) = -CLIP(g(I, Z^T) \odot I, t), \tag{1}$$

where $CLIP(J, s)$ denotes the CLIP matching score between an image $J$ and the text $s$.

The background loss helps ensure that the complementary part to that returned by $g(I, Z^T)$ is irrelevant, in the sense that its CLIP matching score with the text is low.

$$L_{back}(I, t) = CLIP((1 - g(I, Z^T)) \odot I, t), \tag{2}$$

The relevancy map $H$ obtained by the method of Chefer et al. [14] on $CLIP(I, t)$ provides us with another important cue. While it is not entirely reliable in the sense that irrelevant regions are often marked and those regions often do not include the entire relevant part, we find that it often highlights some of the relevant parts. Since both $H$ and $g(I, Z^T)$ have the same range of values ([0,1]), we employ the squared Euclidean norm of the difference.

$$L_{rmap}(I, H) = \|H - g(I, Z^T)\|^2 \tag{3}$$

The background loss may encourage $g(I, Z^T)$ to be as inclusive as possible, so that the complement is minimal. Additionally, the relevancy map loss $L_{rmap}$ may encourage the obtained foreground mask to be overly inclusive. To encourage $g$ to be more spatially limited, we add a regularization loss

$$L_{reg}(I, g)) = \|g(I, Z^T)\| \tag{4}$$

The combined loss is defined as $L = \lambda_1 * L_{fore}(I, t) + \lambda_2 * L_{back}(I, t) + \lambda_3 * L_{rmap}(I, H) + \lambda_4 * L_{reg}(I)$, where $\lambda_1, ..., \lambda_4$ are fixed weighting parameters for all datasets, which were determined after a limited hyperparameters search on the CUB [76] validation set to be 1, 1, 4, 1 respectively.

### 3.1 Inference Tasks

**Weakly supervised object localization (WSOL)** The goal of the WSOL task is to localize a single object in the image $I$ without local annotation. For each fine-grained dataset, there is a single type of object (birds, cars, dogs, etc), and we do not guide the segmentation process with any type of conditioning. The text is used only in the training time for the loss functions, and is simply the name of the object ("birds", "cars", "dogs").

For converting the obtained continuous segmentation map $g(I)$, which is a sigmoid output between 0 to 1, to a binary mask, we follow the binarization used, e.g., by Qin et al. [59, 18, 17, 66], which is the standard in the benchmark. Namely, low-value pixels, below the threshold of 0.1, are zeroed out. Then, the method of Suzuki et al. [70] is used to extract contours. From the largest contour found, the algorithm selects the bounding box of the object.

**Weakly supervised phrase-grounding (WSG)** The aim of weakly supervised phrase-grounding is to generate a localization map, given an image $I$ and a textual phrase $t$ for a specific object in the image. To extract the objects from the output map of $M = g(I, Z^T)$, we first zero low-value pixels, using a threshold of 0.5. Next, we find contours with the method of Suzuki et al. [70]. For each contour, we extract the proper bounding box. The score for each box is calculated from the output map in the positions of the detected box. In the final stage, we apply non-maximal suppression over the boxes with 0.3 IoU, and filtering boxes with a low score (50% less) compared to the maximum score. See Appendix D for the visualization of these steps.

**WWbL Inference** The WWbL task is an open-world localization task, in which the only input is a given image. The task consists of localizing and describing all the elements composing the scene. To solve this task, we propose a two-stage algorithm based on a selective search [74] algorithm to extract object proposals to describe local regions in the image and incorporate the learned network $g$, the image captioning network BLIP, and the CLIP model.

Captioning models usually tend to describe the main element of the scene, neglecting other objects or the background. To address this issue, our proposed algorithm extracts local regions from the image with the Selective Search [74] procedure, which first over-segments the image into super-pixels, and then groups those in a bottom-up manner to propose candidate bounding boxes, see line 2 of Alg. 1. Each region is passed through BLIP to generate a local caption (line 3).

To avoid duplicate captions, the algorithm projects the captions to CLIP space (line 4) and applies the Community Detection (Cd) algorithm [9] to find caption clusters (line 5). The clustering algorithm uses cosine similarity in CLIP space between each caption such that a matrix of size $M \times M$ is obtained, where M is the number of captions. The algorithm employs two hyperparameters: (i) cosine similarity threshold for items in the same cluster and (ii) minimum size of a cluster such that, for each caption (row in the matrix), we find the number of captions above the threshold and check whether it is above the minimum size for a cluster. The final stage of Cd filters out overlapping clusters, giving priority to larger clusters. We set the cosine similarity threshold to be 0.85, and the minimum size to

**Algorithm 1:** WWbL inference method

**Require:** Input image $I$
 1: Load trained networks $g$, BLIP and $E_t$
 2: $\{P_i\}_{i=1}^n = SelectiveSearch(I)$
 3: $\{T_i\}_{i=1}^n = BLIP(\{P_i\}_{i=1}^n)$
 4: $\{Z_i^T\}_{i=1}^n = E_t(\{T_i\}_{i=1}^n)$
 5: $\{T_i, Z_i^T\}_{i=1}^m = Cd(\{Z_i^T\}_{i=1}^n, 2, 0.85)$
 6: $D \leftarrow \phi$
 7: **for** $i = 1....m$ **do**
 8:     $M_i = g(I, Z_i)$          ▷ Generate map
 9:     $B_i \leftarrow BE(M_i)$  ▷ Extract bounding box
10:     $D = D \cup (B_i, T_i)$          ▷ Add object
11: return D

**Figure 3:** Sample WWbL results for Filckr30K

be 2. This algorithm returns a caption for each cluster and its embedding in CLIP space. For each cluster, $g$ generates a localization map. Next, the bounding boxes are extracted like in Sec.3.1.

## 3.2   Architecture

Our network $g$ is used in two types of tasks where (1) image captioning is used only for the loss function (WSOL) and $g$ receives only the image (2) image captioning is used as input to $g$ and in the loss function (WSG and WWbL). In this section, we describe each mode.

**WSOL Architecture**   For WSOL, Network $g$ is based on a U-Net [62] architecture, with skip connections between the encoder and the decoder for signals with the same spatial resolution. The decoder of $g$ employs five upsampling layers, with each block containing two convolutional layers with batch normalization after the last convolutional layer before the activation function. The last activation function is Sigmoid, while the others are Relu.

**Multi-Model Architecture**   For WSG and WWbL, Network $g$ is based on an encoder-decoder architecture adapted to support text-based conditioning. This is illustrated in Fig. 2.

The encoding $Z_t$ for the input text $t$ is produced by the CLIP text encoder, where the size of the vector $Z_t$ is 512. The image encoder of $g$ is a VGG network [68], in which the receptive field is of size $16 \times 16$, i.e., the image is downscaled four times. The number of channels of the obtained map $Z_I$ is also 512. The VGG network is initialized using an ImageNet pre-trained model.

We then consider the vector in $\mathbb{R}^{512}$ associated with each spatial location of the tensor $Z_I$. It is normalized to have a norm of one. The dot product with the vector $Z_t$ (which in CLIP is also normalized) is computed. Performing this over all spatial locations, we obtain a map $Z_s$ with values between -1 and 1. $Z_s$ has a single channel and the same spatial dimensions as $Z_I$.

$Z_s$ can be viewed as a text-conditioned importance map. All channels of $Z_I$ are multiplied by $Z_s$ and the result is passed to the decoder part of $g$. This decoder consists of three upsampling blocks, each with a sampling factor of two. Each block contains two convolutional layers with a kernel size equal to 3 and zero padding equal to one. Batch normalization is used after the last convolution layer, before the activation function. The first layer's activation function is a ReLU, while the last layer's activation function is a Sigmoid.

## 4   Experiments

We present our results for the three tasks: (i) weakly supervised object localization (WSOL) , (ii) weakly supervised phrase grounding (WSG), with training on either MSCOCO 2014 [43] or the Visual Genome (VG) dataset [40], and (iii) the new task we present (WWbL). For the first task, we employ three fine-grained localization datasets, and for the other two, we use the three datasets commonly used in WSG.

**Datasets**   For the task of WSOL we evaluate our model on fine-grained datasets often used for this task. CUB-200-2011 [76] contains 200 birds species, with 11,788 images divided into 5994 training images and 5794 test images. Stanford Car [39] contains 196 categories of cars, with 8144 samples

**Table 2:** WSOL results for the CUB dataset.

| Method | Backbone | Accuracy[%] |
|---|---|---|
| GAE [12] | CLIP | 68.01 |
| CAM [92] | VGG16 | 55.10 |
| ACoL [90] | VGG16 | 62.96 |
| ADL [19] | VGG16 | 75.41 |
| DANet [84] | VGG16 | 67.70 |
| MEIL [48] | VGG16 | 73.84 |
| GCNet [46] | VGG16 | 81.10 |
| POSL [87] | VGG16 | 89.11 |
| SPA [55] | VGG16 | 77.29 |
| SLT [28] | VGG16 | 87.60 |
| FAM [51] | VGG16 | 89.26 |
| ORNet [82] | VGG16 | 86.20 |
| BAS [80] | VGG16 | 91.07 |
| Ours | VGG16 | **93.94** |
| CAM [92] | MobileNetV1 | 63.30 |
| HaS [69] | MobileNetV1 | 67.31 |
| FAM [51] | MobileNetV1 | 85.71 |
| RCAM [6] | MobileNetV1 | 78.60 |
| BAS [80] | MobileNetV1 | 92.35 |
| Ours | MobileNetV1 | **94.40** |
| CAM [92] | Resnet50 | 57.35 |
| POSL [87] | Resnet50 | 90.00 |
| WTL [4] | Resnet50 | 77.35 |
| FAM [51] | Resnet50 | 85.73 |
| SPOL [78] | Resnet50 | 96.46 |
| BAS [80] | Resnet50 | 95.13 |
| Ours | Resnet50 | **96.54** |
| CAM [92] | InceptionV3 | 55.10 |
| DANet [84] | InceptionV3 | 67.03 |
| I2C [91] | InceptionV3 | 72.60 |
| GCNet [46] | InceptionV3 | 75.30 |
| SPA [55] | InceptionV3 | 72.14 |
| SLT [28] | InceptionV3 | 86.50 |
| FAM [51] | InceptionV3 | 87.25 |
| BAS [80] | InceptionV3 | 92.24 |
| Ours | InceptionV3 | **94.30** |
| PSOL [88] | DenseNet161 | 93.01 |
| Ours | DenseNet161 | **93.71** |

**Table 3:** Localization accuracy (percentage) WSOL on the cars and dogs datasets.

| Method | Stanford-Cars | Stanford-Dogs |
|---|---|---|
| CLIP [12] | 81.2 | 64.7 |
| CAM [92] | 65.2 | 66.0 |
| HaS [92] | 87.4 | 77.5 |
| ADL [19] | 82.8 | 73.5 |
| RDAP [17] | 92.9 | 77.7 |
| FG [66] | 96.2 | 79.2 |
| Ours | **98.9** | **86.4** |

**Table 4:** Backbones comparison for weakly supervised segmentation, as evaluated on CUB

| Backbone | Params[#] | Accuracy[%] |
|---|---|---|
| Resnet18 | 11.1M | 94.85 |
| Resnet34 | 21.3M | 95.31 |
| Resnet50 | 23.5M | 96.54 |
| Resnet101 | 42.5M | 96.64 |
| Hardnet39DS | 3.5M | 93.77 |
| Hardnet68DS | 4.2M | 93.85 |
| Hardnet68 | 17.6M | 94.99 |
| Hardnet85 | 36.7M | 95.11 |

**Table 5:** Ablation results for weakly supervised localization on CUB[76].

| $R(g(I))$ | $L_{rmap}$ | $L_{fore}$ | $L_{back}$ | Accuracy[%] |
|---|---|---|---|---|
| - | - | - | ✓ | 48.36 |
| - | - | ✓ | - | 47.58 |
| - | - | ✓ | ✓ | 42.11 |
| ✓ | - | - | ✓ | 69.01 |
| ✓ | - | ✓ | - | 84.72 |
| ✓ | - | ✓ | ✓ | 90.02 |
| - | ✓ | - | - | 87.02 |
| - | ✓ | - | ✓ | 89.12 |
| - | ✓ | ✓ | - | 90.09 |
| - | ✓ | ✓ | ✓ | 91.08 |
| ✓ | ✓ | - | - | 84.24 |
| ✓ | ✓ | - | ✓ | 92.67 |
| ✓ | ✓ | ✓ | - | 94.73 |
| ✓ | ✓ | ✓ | ✓ | 96.54 |

in the training set and 8041 samples in the test set. Stanford dogs [36] consists of 20,580 images, with a split of 12,000 for training and 8580 for testing, where the data has 120 classes of dogs.

For the task of WSG and WWbL, our model trains on the split of MSCOCO and VG, and we evaluate it on the test splits of Flickr30k, ReferIt, and VG. MSCOCO 2014 [43], using the splits of Akbari et al. [2], consists of 82,783 training images and 40,504 validation images. Each image has five captions describing it. VG [40] contains 77,398 training, 5000 validation, and 5000 test images. Each image comes with a set of free-form text annotated bounding boxes.

Flickr30k [58] Entities, which is based on Flickr30k, contains 224K phrases describing objects in over 31K images, each described by 5 captions. For evaluation, we use the same 1k images from the test split of Akbari et al. [2]. ReferIt [27, 15] contains 20K images and 99,535 segmented images that also contain a description for the entire image. These image regions were collected in a two-player game with approximately 130K isolated entity descriptions. We use the split of Akbari et al. [2].

**Implementation details**    In WSOL, our Algorithm receives an input image of size $224 \times 224$. During training, the image is first resized to $256 \times 256$ and then a random crop of size $224 \times 224$ is extracted. During the evaluation, the image is resized to $224 \times 224$.

Network $g$ for WSOL is based on an Imagenet pretrained visual encoder. We compare the performance obtained using VGG16 [68], MobileNetV1 [30], Resnet50 [29], InceptionV3 [71] and

**Table 6:** WSG results: "pointing game" on VG, Flickr30K, and ReferIt.

| Method | Backbone | VG trained | | | MS-COCO trained | | |
|---|---|---|---|---|---|---|---|
| | | VG | Flicker | ReferIt | VG | Flicker | ReferIt |
| Baseline | Random | 11.15 | 27.24 | 24.30 | 11.15 | 27.24 | 24.30 |
| Baseline | Center | 20.55 | 47.40 | 30.30 | 20.55 | 47.40 | 30.30 |
| GAE [12] | CLIP | 54.72 | 72.47 | 56.76 | 54.72 | 72.47 | 56.76 |
| FCVC [23] | VGG | - | - | - | 14.03 | 29.03 | 33.52 |
| VGLS [81] | VGG | - | - | - | 24.40 | - | - |
| TD [89] | Inception-2 | 19.31 | 42.40 | 31.97 | - | - | - |
| SSS [33] | VGG | 30.03 | 49.10 | 39.98 | - | - | - |
| MG [2] | BiLSTM+VGG | 50.18 | 57.91 | 62.76 | 46.99 | 53.29 | 47.89 |
| MG [2] | ELMo+VGG | 48.76 | 60.08 | 60.01 | 47.94 | 61.66 | 47.52 |
| GbS [3] | VGG | 53.40 | 70.48 | 59.44 | 52.00 | 72.60 | 56.10 |
| ours | CLIP+VGG | **62.31** | **75.63** | **65.95** | **59.09** | **75.43** | **61.03** |

**Table 7:** WWbL and WSG results: "pointing game" and bounding box accuracy.

| Task | Model | Training | Test Point Accuracy | | | Test Bbox Accuracy | | |
|---|---|---|---|---|---|---|---|---|
| | | | VG | Flickr | ReferIt | VG | Flickr | ReferIt |
| WWbL | MG[2] | COCO | 32.91 | 50.12 | 36.34 | 11.48 | 23.75 | 13.31 |
| | MG[2] | VG | 32.15 | 49.48 | 38.06 | 12.23 | 24.79 | 16.43 |
| | GAE [12] | - | 38.15 | 56.25 | 41.64 | 9.69 | 17.14 | 12.31 |
| | ours | COCO | **44.20** | **61.38** | 43.77 | 17.76 | **32.44** | **21.76** |
| | ours | VG | 43.91 | 58.59 | **44.89** | **17.77** | 31.46 | 18.89 |
| WSG | MG [2] | COCO | 47.94 | 61.66 | 47.52 | 15.77 | 27.06 | 15.15 |
| | MG [2] | VG | 48.76 | 60.08 | 60.01 | 14.45 | 27.78 | 18.85 |
| | GAE [12] | - | 54.72 | 72.47 | 56.76 | 16.70 | 25.56 | 19.10 |
| | ours | COCO | 59.09 | 75.43 | 61.03 | 27.22 | 35.75 | 30.08 |
| | ours | VG | **62.31** | **75.63** | **65.95** | **27.26** | **36.35** | **32.25** |

DenseNet161 [31] is applied for CUB dataset to compare performances with the state-of-the-art baselines, while for the other datasets we used Resnet50, since that is what the literature baselines have published. A SGD optimizer with a batch size of 48 and an initial learning rate of 0.0003 for 100 epochs is used. The optimizer momentum of 0.9 and weight decay of 0.0001 are also used. During the training, a random horizontal flip with 0.5 probability is applied. All models are trained on a single GeForce RTX 2080Ti Nvidia GPU. In WSG training (the same network is used for inference in both WSG and WWbL), following Akbari et al. [2], network $g$ receives an input image of size $299 \times 299$ and generates a localization map of the same size. During training, the image is also resized to $224 \times 224$, which is the input size of CLIP. The relevancy map $H$ is generated at this resolution and is resized to $299 \times 299$. Network $g$ for WSG and WWbL is VGG16 [68], to ensure a fair comparison with the baselines. An SGD optimizer (batch size of 32 and an initial learning rate of 0.0003) is used for 100 epochs. The optimizer momentum of 0.9 and weight decay of 0.0001 are also used. During training, a random horizontal flip is applied, with 0.5 probability. All models are trained on a double 2080Ti Nvidia GPU, while all experiments of Arbelle et al. [3] were conducted on four V100 GPUs.

**Results** For the WSOL task, the main accuracy metric measures whether the intersection over union (IoU) of the ground-truth bounding box and the detector's output are above 0.5. As can be seen in Tab. 2, our method obtains state-of-the-art performance for the CUB [76] dataset, for all five different backbones. The method also outperforms GAE [12] with CLIP as a backbone. Our method improves the GAE map, which it uses during training, by more than 25% in the accuracy metric. The results listed in Tab. 3 show that our method also achieves state-of-the-art performance for the fine-grained localization datasets. Following previous work, a Resnet-50 is used for these methods, except for GAE, which is applied with a CLIP backbone. Appendix E presents sample results.

For the WSG task, the algorithm is evaluated with respect to the accuracy of the pointing game [89], which is calculated from the output map by finding the maximum-value location for the given query and checking whether this point is located in the region of the object. Another metric we report (B-Box accuracy) compares the extracted bounding-boxes with the bounding-box annotations in the same manner as the WSOL task above. For a fair comparison, we use the same training/validation/test splits as Akbari et al. [2]. For ReferIt, Visual Genome and Flickr30K, we treat each query as a single sentence. Tab. 6 summarises the results for the Flickr30k, ReferIt and Visual Genome datasets for the WSG task. Evidently, our method is superior to all baselines, whether training takes place over VG or MS-COCO. Tab. 7 presents bounding box accuracy results for the WSG task. Here, too, our method outperforms the baseline methods. In appendix B we present sample output maps.

For the WWbL task, we use the same metrics (pointing game and bounding-box accuracy). For each ground-truth pair of bounding box and caption, we select the closest caption in CLIP space from the output predictions and compare the output map to the bounding box using the pointing accuracy metric. In addition, bounding boxes are extracted for the heat-map, as described at Sec. 3.1, and compared to the selected ground-truth bounding boxes with the same accuracy metric as above.

Tab.7 presents the results for the WWbL task. In addition to our method (Alg. 1) that utilizes network $g$ to generate a heat-map, given an image and a text (line 8), we also test two alternative heatmap generation methods. The first one is the MG method of Akbari et al. [2] for the WSG task, and the second is the GAE explainability method [12]. As can be seen in the results listed in Tab. 7, our proposed method obtains the best results among the three. For comparison, we also provide WSG results using Alg. 1 and the ground truth text for the same three heatmap methods. Interestingly,

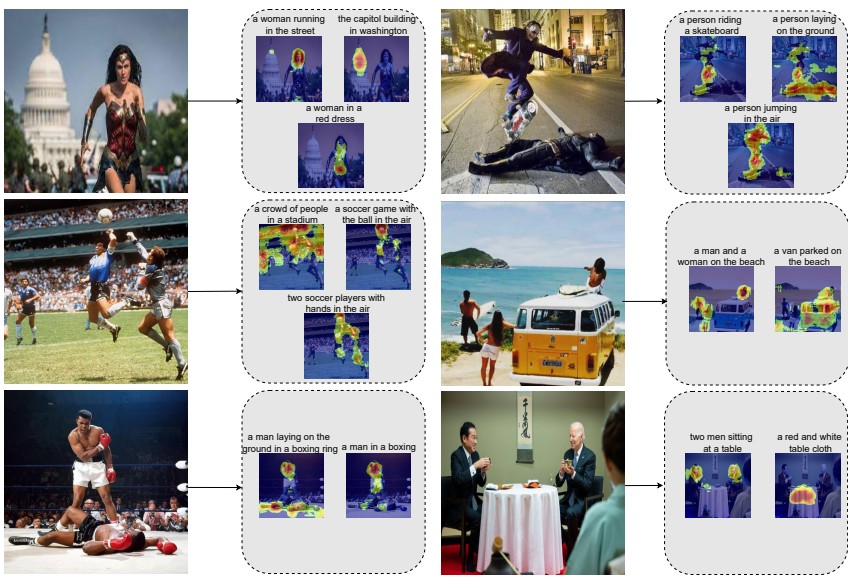

**Figure 4:** WWbL on a few samples of web images.

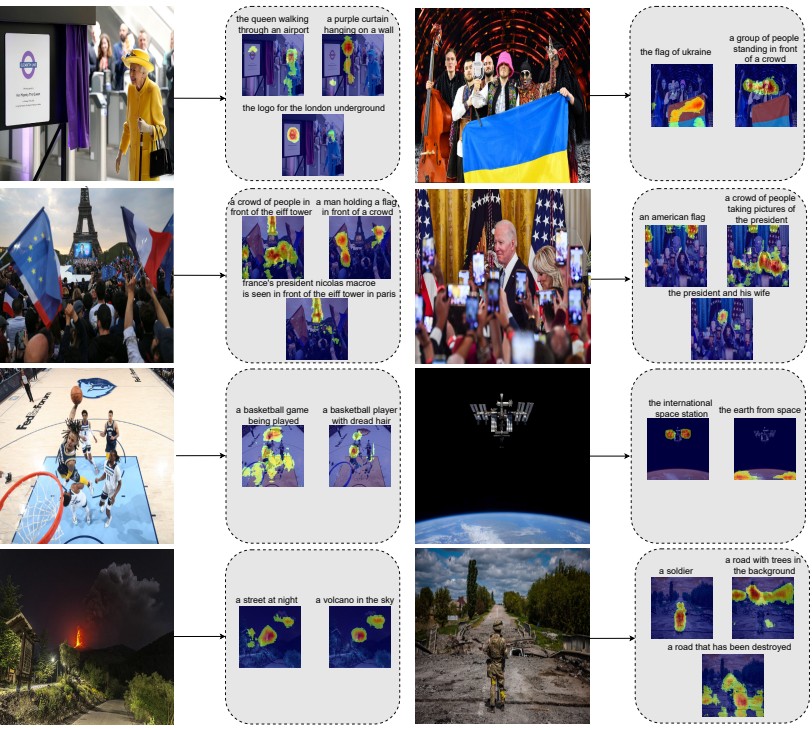

**Figure 5:** WWbL on images from Week in pictures by BBC (April-May 2022).

the $g$ that is trained on the VG dataset shows better performance for WSG, while in WWbL the results are mixed. Fig 3, Fig. 4 show results for our method from Flickr30K and web images samples respectively. Fig. 5 shows results for our method from BBC - Week in pictures from the recent weeks (April-May 2022).

**Ablation Study**    In this section, we compare the accuracy of $g$ with different modes of the loss function and present the experiment used to determine the values of $\lambda_{1,2,3,4}$. In addition, we also

compare the performance of two backbone families, Resnet [29] and HarDnet [10]. All experiments were conducted in the WSOL settings for the CUB[76] dataset. Tab. 5 presents the sensitivity of our network $g$ to each loss term, as evaluated on the test set, using the Resnet50 backbone. The first part of the table compares the performance without $L_r map$, using different combinations of the other loss terms. As can be seen, the best performance is achieved when we use the three terms $L_{fore}, L_{back}, L_{reg}$ together. Each term improves the performance: $L_{fore}$ improves it by 21%, $L_{back}$ by 5% and $L_{reg}$ by 48%. The second part compares the performance with $L_{rmap}$ which improved by more than 6%. As can be seen, each of the other terms improves the performance: $L_{fore}$ improves it by 4%, $L_{back}$ by 2%, and $L_{reg}$ by 5.5%. Also, training only with $L_{rmap}$ decreases the performance by more than 9% compared to performance using all loss terms. The backbone ablation was conducted with two backbone families and is summarised in Tab. 4. For both families, performance improves as the number of parameters is increased.

Appendix C, presents the sensitivity of the network $g$ to the $\lambda_3$ coefficient, which is the weight of $L_{rmap}$ and the only coefficient that is not set to 1 in our experiments. The method seems to be largely insensitive to this parameter and it is stable for a wide range of values. The ability of $g$ to generalize well under different pre-trained CLIP models is examined in App. G. Evidently, the variance in the performance of $g$ is much lower than the variance in the zero-shot classification performance of the CLIP models used to train $g$.

## 5 Discussion and limitations

When biological systems see, the input is seldom limited to single images. Although our WWbL method was developed in the context of stills, it can be naturally extended to perform online localization of objects in video. For this, the detected regions and their matching concepts need to be tracked and must evolve over time, with the ability to correct previous hypotheses. We leave this effort for future research, since benchmarking this ability with the current datasets is challenging.

The common computer vision terminology makes the distinction between object localization and object detection. The former aims to locate the main object of a certain type in the input image, while the latter aims to find all the relevant objects and their individual boundaries. Our WWbL method aims to extract every existing object and can, therefore, detect all image objects.

The method employs the selective search method [74]. This can be avoided by identifying one region at a time and iterating the process. Such a method is presented in Appendix A. It uses $g$ to recover the object map at each stage. The results reported in this Appendix are somewhat lower than the method based on a selective search. Since the image datasets and pre-trained models our method employs may be biased towards western media concepts, it is culturally biased. For similar reasons, it may display discriminatory behavior.

Another limitation is that our weakly supervised learning scheme does not distinguish between multiple instances of the same object. While Algorithm 1 can be improved to somewhat mitigate this, by separating multiple objects that have the same caption, building such a solution robustly may be challenging without additional supervision. App.F present visualization for this scenario.

## 6 Conclusions

Through the power of large pre-trained transformers and by integrating explainability cues, we build an effective weakly supervised phrase-grounding network $g$. By combining an off-the-shelf image captioning engine, we are able to identify and localize the objects within an input image. The task we solve is an open-world one and the setting generalizes multiple existing tasks. It is convincingly demonstrated that our weakly supervised solution is on par with the fully supervised alternatives that exist for the phrase-grounding task.

## Acknowledgments

This project has received funding from the European Research Council (ERC) under the European Unions Horizon 2020 research and innovation programme (grant ERC CoG 725974).

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
