# OpenReview forum: "What is Where by Looking: Weakly-Supervised Open-World Phrase-Grounding without Text Inputs"
_NeurIPS.cc/2022/Conference — NeurIPS 2022 Accept_

### Official Review · Reviewer_MVej · 2022-07-09

**Rating:** 4
**Confidence:** 3
**Soundness:** 2 fair
**Presentation:** 3 good
**Contribution:** 3 good

**Summary:**

This paper proposes a new solution for weakly-supervised open-world phrase grounding. The method is built upon the off-the-shelf CLIP model and captioning model, only taking images as the input. The experiments are conducted on several datasets, showing the superior performance of the proposed solution.

**Questions:**

1. Have the authors tried to change the CLIP model and BLIP to other models to see how the new alternative can perform?

2. Have the authors compare the model with some pre-trained methods like CPT-Colorful Prompt Tuning, Frozen or Flamingo?

**Limitations:**

How can the proposed solution perform if there are no strong pre-trained models available?

**Strengths And Weaknesses:**

Strengths:
1. The method is simple yet effective. By combining strong pretrained model and off-the-shelf model, it achieves good performance without text inputs.

2. The empirical results are several benchmarks are promising.

Weaknesses:
1. The novelty of the paper is quite limited. Most of the components are not developed by the paper.

2. It is not clear how this method can perform well in such a weakly-supervised setting. Its generalization ability may be contributed by the powerful pretrained model (CLIP). The author should performant some ablation study to change these models to other candidates to validate this.

3. The paper ignored some work related to pretrained model for zero/few shot settings (with prompt, etc) [a,b]
a. A Good Prompt Is Worth Millions of Parameters: Low-resource Prompt-based Learning for Vision-Language Models. ACL 2022.
b. CPT: Colorful Prompt Tuning for Pre-trained Vision-Language Models
c. SimVLM: Simple Visual Language Model Pretraining with Weak Supervision. ICLR 2022.

---

> ### Author Response · Authors · 2022-08-02
> **Thank you for your detailed comments.**
>
> Our paper focuses on two main contributions: first, a new state-of-the-art architecture for weakly-supervised localization with text and image as input. Second, building upon the first contribution, we tackle a purely-visual task called WWbL, in which given an input image and nothing else, an algorithm should predict what are objects in the image and where. This is the most fundamental task in computer vision, yet it was not dealt with in previous work. This is an open-world detection task that does not rely on any localization information during training.
>
> We respectfully disagree that there is a limited novelty. The weakly-supervised trained localization network and the ability to solve the task are completely novel. The first employs new loss terms and techniques, and the second has no precedents, despite being specifically identified decades ago as the most fundamental problem in computational vision (Marr, 1980). The fact that there are no precedents points to the level of the challenge. The list of requirements in the WWbL problem is beyond that of any localization method: the weak supervision, the open-world requirements, and the need to provide a full textual description.
>
> It is true that our method builds upon existing models, including CLIP, BLIP, and selective search. However, the ability to combine these relies on a crucial new component, which is the new weakly-supervised localization method. The utilization of large image-text models, such as CLIP and BLIP, for other tasks than the ones used for training these,  is a very active research domain [L27-34]. This points to the validity of our research direction: building upon such models, we are able to solve, for the first time, the most fundamental computational vision problem.
>
> We acknowledge that our method’s success is conditioned on the quality of the image-text models it utilizes. This is, of course, true for most work that builds upon pre-trained models. Following the review, we examine the ability of network $g$ to generalize when switching between various CLIP models. As can be seen in Appendix H, the variance in the performance of $g$ is much lower than the variance in the zero-shot classification performance of the CLIP models used to train $g$.
>
> A similar experiment was conducted for the phrase grounding task. Due to resource constraints up to this point in time, we are only able to do it with the RN50 backbone trained on the open-ai dataset. The results are extremely encouraging and show only a modest drop in the performance compared to the CLIP model released by open-ai (75.1 vs 75.6 on flicker30K). This gap is much narrower than the gap that we see in zero-shot classification and in the WSOL task on the CUB benchmark.
>
> Thank you for pointing us to three recent related works. We have added these to the related work section. Contributions (a,c) solve vision-language tasks such as image captioning and VQA. These can be used in our work instead of BLIP, but this effort is out of the scope of our current effort.  Work (b) is a phrase grounding work, formulated as masked language modeling. It is supervised, and our work is weakly supervised.
>
> We are not sure we understand the request “Have the authors compare the model with some pre-trained methods like CPT-Colorful Prompt Tuning, Frozen, or Flamingo?” CPT-colorful is a phrase grounding model that uses a fully supervised training scheme, while our work does not use any location annotation. Flamingo is a captioning model that is especially suited for few-shot learning. We could, in principle, adopt our method for using it instead of BLIP. However, this is out of the scope of the current effort and, in any case, the code and weights of Flamingo are not public.

---

> > ### Author Response · Authors · 2022-08-08
> > **Additional experiment - Appendix H**
> >
> > We have added an experiment that tests the accuracy of $g$ with different pre-trained CLIP models for the Phrase grounding task. The results, which can be found in Appendix H of the revised manuscript, indicate that the ResNet50 model slightly outperforms the ViT32 one when both are trained on the Open-AI dataset. Other datasets, both bigger and smaller, do not lead to better performance.
> >
> > As mentioned before, we note that the gap observed for both WSOL and Phrase Grounding is much narrower than the gap that we see in top-1 accuracy for zero-shot classification. This indicates that our method is relatively stable concerning the CLIP model used during training.

---

> > ### Comment · Reviewer_MVej · 2022-08-09
> > **Reply of by Reviewer MVej**
> >
> > Thank you for the authors' reply.
> >
> > The major concern of mine is still the novelty. As also pointed out by other reviewers, is a direct combination of image-caption (region-based) + localization models. This level of novelty does not meet the bar of a NeurIPS paper. The authors also claimed that another contribution of the paper is proposing a new purely-visual task WWbL. However, in my opinion, the style of the paper is not a task-oriented paper, which should focus on benchmarks construction, and description of the motivation of the new task, etc.
> >
> > In addition, my question on how the CLIP and BLIP models can affect the empirical results has not been well addressed. I would like to see replacements of CLIP and BLIP with other models, not different checkpoints of CLIP.
> >
> > I will keep my original rating.

---

> > > ### Author Response · Authors · 2022-08-09
> > > **Further clarifications**
> > >
> > > We thank the reviewer for engaging in this discussion.
> > >
> > > __Re: Novelty__
> > >
> > > The novelty argument made by the reviewer is based on a very high-level summary of our work and disregards our actual novelty claims. These claims are:
> > > 1. We are the first to train a wealy supervised phrase grounding model in a way that utilizes pre-trained visual-language models.
> > > 2. The training of this model is based on a novel architecture and novel loss terms, namely $L_{fore}$ and $L_{back}$.
> > >
> > > Since the ability in item 1 is useful (as Tab. 6 indicates) and since it is the basis of being able to solve WWbL, hopefully, we can agree that it is important enough. A counterargument to our novelty claims should be, therefore, in the form of identifying a previous work doing either 1 or 2.
> > >
> > > Regarding WWbL: to be precise, our claims are:
> > > 1. It (weakly supervised trained, no text input at inference, output are a set of descriptions and segmentation maps) is a fundamental computer vision task, which was identified as such in the early days of vision research.
> > > 2. It was never solved before.
> > >
> > > We can probably agree on both points.
> > >
> > > __Re: Alternative models__
> > >
> > >
> > > Regarding alternative image-language models: following the review, we provided 8 different alternatives to the original CLIP we used. Unfortunately, we did not provide the alternative the reviewer wished for. This will be promptly corrected. However, is it fair to ignore the effort that was already done and the very encouraging conclusions? One surprising conclusion is that the method is much less sensitive to the underlying model than the simple zero-shot classification method.

---

### Official Review · Reviewer_aWBt · 2022-07-10

**Rating:** 8
**Confidence:** 4
**Soundness:** 4 excellent
**Presentation:** 4 excellent
**Contribution:** 4 excellent

**Summary:**

In this paper, the authors explore using pre-trained CLIP for three tasks: (1) object localization, (2) phrase grounding and (3) WWbL: generating object masks and corresponding captions for an image. Their model encodes both image and text with VGG and CLIP respectively, and generates a heat map. They use two weakly supervised losses: (1) foreground loss which encourages the masked image with the heat map to match the text input, (2) background loss which encourages the remaining image to not match the text input, (3) relevancy heatmap which measures the difference between the generated heat map and the relevancy heatmap of CLIP, and (4) regularization loss for the generated heat map. Performing inference on (1) and (2) are trivial, whereas for (3) they designed an algorithm which first unsupervisedly propose region, and then get captions for them to convert the problem to a phrase grounding task.

Object localization results on CUB, and phrase ground results on VG, Flickr and ReferIt show that their model is better than previous methods, and each of the four losses plays a role in improving the results. On their proposed new task, WWbL, the results are also in favor of their proposed model. Combining all of the results, this paper found that CLIP can be useful in various language related object segmentation tasks.


**Questions:**

Please refer to the weakness section above.

**Limitations:**

It is nice of the authors to point out the limitation that their work only applies to single images. It would also be nice if the authors can also talk about the potential ethical impact of their paper, e.g. would the bias in the pre-trained models be reinforced or reduced in this method?

**Strengths And Weaknesses:**

Strengths:

It is natural to use a pre-trained vision-language model for object detection/localization. Straightforward and simple solutions like [1], have been proposed and could perform reasonably well in a zero-shot manner. This paper is also a neat method, based on the assumption that objects to localize have higher similarity to the text in CLIP embeddings, compared to background. With the four simple and intuitive losses, the authors show their model performs better than the previous methods.
The authors also contribute a new task, WWbL.

[1] https://github.com/shonenkov/CLIP-ODS

Weaknesses:

1. The model may fail for ambiguous captions. For example, given an image of four apples, if the text is “an apple”, the desired output mask should be one of the apples for a deterministic model. Since the foreground loss and background loss maximizes the difference between the image representation along the text embedding dimension, one of the most likely results is that all of the apples are in the foreground (if CLIP “thinks” four apples to no apples is more different than one apple and three in terms of the caption “an apple”). This means that even for a perfect pre-trained model, such loss may still result in undesired output given ambiguous captions.

2. Related to 1, this paper lacks a qualitative study of the effects of coefficient. It is shown in this paper that all of the losses are important for the metrics, but would different configurations have different tendencies? Can configurations be optimized for different applications? This might be out of scope, but if the authors had provided insights on this, this paper would have been much stronger.

---

> ### Author Response · Authors · 2022-08-02
> **Thank you very much for the very supportive review and for the specific feedback.**
>
> We now note in the limitations section that “our weakly supervised learning scheme does not distinguish between multiple instances of the same object. While Algorithm 1 can be improved to somewhat mitigate this, by separating multiple objects that have the same caption, building such a solution in a robust way may be challenging without additional supervision.”
>
> In the new appendix G, we present our method's results for images with multiply instances of the same object, such as apples and dogs. We present both the results of the WWbL method and the results for grounding specific sentences that are used to study the output of network $g$.
>
> The results indicate that WWbL does not typically select sentences that distinguish between the various objects. However, network $g$ has the capacity to separate between different objects of the same class given specific captions. When there are multiple objects of exactly the same type, e.g., multiple green apples, $g$ marks all of them. We note that $g$'s heatmap does peak at specific parts of the objects, which may facilitate instance separation. Both the ability to extract specific captions automatically and the ability to perform instance segmentation are left for future work.
>
> Following the review, we have also applied our method with different weighting parameters. Due to the time limit, we have performed this sensitivity analysis only for $\lambda_3$, which is the only coefficient that is not set to one in our experiments (it is fixed at a value of 4). The results are presented in Table10 in appendix C. Evidently, for a wide range of this parameter, the performance of the $g$ is similar. We also note that the same $\lambda_3$ seems to be optimal for different tasks.
>
> Following another review, we have also replaced the pre-trained CLIP model with other models, in order to validate the robustness of our method to a switch of the underlying visual-language model. As can be seen in Appendix H, the variance in the performance of $g$ is much lower than the variance in the zero-shot classification performance of the CLIP models used to train $g$.

---

> ### Comment · Area_Chair_aArC · 2022-08-09
> **Please look at respond to author response**
>
> Dear reviewer aWBt,
>
> please look at the author response to your review, and comment on the corresponding author response, and if this changes your ratings / understanding / resolves your concerns / creates new concerns/questions.
>
> Thank you, your AC
>
> PS: Don't respond to this message but directly to the author response, so author can also see your response!

---

### Official Review · Reviewer_J9Sc · 2022-07-11

**Rating:** 4
**Confidence:** 3
**Soundness:** 2 fair
**Presentation:** 1 poor
**Contribution:** 2 fair

**Summary:**

This paper proposes and studies a new task, which localizes image regions with masks and describes them with natural language. The paper studies the task in a weakly-supervised open-world setting. In order to tackle the task, two recently proposed pre-trained vision-language models, namely, CLIP and BLIP are leveraged to produce multi-modal matching score and generate candidate captions, respectively. Meanwhile, an encoder-decoder network is trained to generate foreground mask for grounding. Superior performances are shown in multiple benchmark datasets, for the three evaluation tasks.

**Questions:**

1. Discuss the novelty and elaborate the contributions (Concern #1).
2. Discuss the empirical fairness (Concern #2).

**Limitations:**

The authors have adequately addressed the limitations of their work.

**Strengths And Weaknesses:**

Strengths:

1. A new perspective for vision-language community, and open-world applications. Comprehensive techniques are proposed with extensive supportive experiments.
2. State-of-the-art quantitative results on multiple benchmark datasets for the three tasks (WSOL, WSG, WWbL).

Concerns:

1. Regarding novelty. The novelty might not reach the standard of NeurIPS conference. The proposed framework largely depends on CLIP-like pre-trained models and lacks original innovation. Moreover, the proposal claims to be "purely visual" (without textual input), but actually the textual information comes from pre-trained BLIP model at inference. I expect more discussion regarding the novelty.
2. Regarding empirical results. I am not quite sure whether it is fair to compare with previous methods, considering the proposed method actually leverages the pre-trained information from CLIP and BLIP. And I am also not sure if the datasets (e.g., Flicker, ReferIt) used for comparison have overlapping (or similar) images with the large-scale datasets involved in the pre-trained models (i.e., CLIP and BLIP). The comparison fairness should be discussed.
3. The writing quality and presentation should be improved. The writing and expression highly affect deeper understanding of this work. There are quite many expression issues and typos throughout the whole paper. Some of the typos are listed as follows.
(1) L50, "A two-stage inference time procedure" shoud be "A two-stage inference procedure". (2) L69, "during the inference test" should be "during the inference". (3) L197, "a encoder-decoder architecture" should be "an encoder-decoder architecture". (4) L241, "An SGD optimizer" should be "A SGD optimizer".

---

> ### Author Response · Authors · 2022-08-02
> **Thank you for your detailed comments**
>
> Our paper focuses on two main contributions: first, a new state-of-the-art architecture for weakly-supervised localization with text and image as input. Second, building upon the first contribution, we tackle a purely-visual task called WWbL, in which given an input image and nothing else, an algorithm should predict what objects are in the image and where. This is the most fundamental task in computer vision, yet it was not dealt with in previous work. This is an open-world detection task that does not rely on any localization information during training.
>
> We respectfully disagree that there is a limited novelty. The weakly-supervised trained localization network and the ability to solve the task are completely novel. The first employs new loss terms and techniques, and the second has no precedents, despite being specifically identified decades ago as the most fundamental problem in computational vision (Marr, 1980). The fact that there are no precedents points to the level of the challenge. The list of requirements in the WWbL problem is beyond that of any localization method: the weak supervision, the open-world requirements, and the need to provide a full textual description.
>
> It is true that our method builds upon existing models, including CLIP, BLIP, and selective search. However, the ability to combine these relies on a crucial new component, which is the new weakly-supervised localization method.
>
> "Purely visual" referees to inference time, and not to train time. This would be clarified further. Obviously, one cannot learn such NLP-heavy tasks without any text being involved during training.
>
> As far as we can ascertain, CLIP and BLIP were not trained on the test sets of Flicker or ReferIt used for the evaluation of our work against previous works. To alleviate such concerns, we have also provided qualitative results on datasets of very recent news images (Appendix D Figure 6 and 7).
>
> We compare our method to GAE [12] on both the WSG and WWbL tasks. GAE leverages per-trained information from both CLIP and BLIP. Our method’s results on various datasets are higher with a large margin. It is not feasible for us to retrain GbS[3] and MG[2] using larger datasets.
>
> We note that the utilization of these large image-text models for other tasks is a very active research domain (lines 29-32). This points to the validity of our research direction: building upon such models, we are able to solve, for the first time, the most fundamental computational vision problem.
>
> All requests for elucidation have been fully addressed in the revised version. We would further submit the paper for another round of proofreading.

---

### Official Review · Reviewer_4wQX · 2022-07-13

**Rating:** 5
**Confidence:** 3
**Soundness:** 3 good
**Presentation:** 2 fair
**Contribution:** 3 good

**Summary:**

The paper proposes the task of Weakly-Supervised Open-World Phrase-Grounding, which seeks to first generate captions for local regions and then ground the caption. Meanwhile, the paper proposes a way to train a mask generator given image and text with only weak supervision (based on CLIP).

**Questions:**

See cons.

**Limitations:**

N/A.

**Strengths And Weaknesses:**

Pros:
+ The paper studies a new task and proposes a sensible baseline to this task.
+ The paper contributes a way to train a mask generator with only weak supervision with decent performance on benchmark (e.g., Flickr30K), which is "somewhat" novel to my knowledge.

Cons:
- The newly proposed task is not particularly novel or challenging. It seems like a direct combination of image-caption (region-based) + localization. I would have liked to see some discussion on the unique challenge this task poses and how solving it might benefit real-world needs or other tasks.
- The method to the newly proposed task is also a simple combination of previously methods: selective search for proposing regions; BLIP for generating the captions given a region; a localization network.
- The paper also proposes a way to train a mask generator given image and text with only weak supervision (based on CLIP). However, the paper does not come with a discussion on the novelty of the proposed method and how it compares to previous weakly-supervised methods for localization. The idea of using a "classification" model's heatmap for localization is not particularly new so I would appreciate a detailed discussion.
- The paper is a bit confusing on its core contribution: it lacks a full discussion on the uniqueness of the newly proposed task if the task if the core contribution; the method of training the mask generator lacks a discussion to prior work so it is also hard to judge the contribution of the method part.

Minor points:

Providing more backgrounds on BLIP would be helpful.

MDETR should be referenced.

GLIP [42] does not use CLIP initialization.

---

> ### Author Response · Authors · 2022-08-02
> **Thank you for your detailed comments**
>
> Our paper focuses on two main contributions: first, a new state-of-the-art architecture for weakly-supervised localization with text and image as input. Second, building upon the first contribution, we tackle a purely-visual task called WWbL, in which given an input image and nothing else, an algorithm should predict what objects are in the image and where. This is the most fundamental task in computer vision, yet it was not dealt with in previous work. This is an open-world detection task that does not rely on any localization information during training.
>
> We respectfully disagree that there is a limited novelty and no challenge. The weakly-supervised trained localization network and the ability to solve the task are completely novel. The first employs new loss terms and techniques, and the second has no precedents, despite being specifically identified decades ago as the most fundamental problem in computational vision (Marr, 1980). The fact that there are no precedents points to the level of the challenge. The list of requirements in the WWbL problem is beyond that of any localization method: the weak supervision, the open-world requirements, and the need to provide a full textual description.
>
> It is true that our method builds upon existing models, including CLIP, BLIP, and selective search. However, the ability to combine these relies on a crucial new component, which is the new weakly-supervised localization method. These points are already made in the paper, but we did not include the list of contributions the reviewer asks for. This is now added to the revised version.
>
> The words “simple combination” used by the reviewer, could be a valid description of algorithm 1 but not of our entire work. To enable algorithm 1, one has to first obtain network g (the localization network), which, as shown in our ablation study, requires multiple novel loss terms.
>
> The question “how it compares to previous weakly-supervised methods for localization” is answered in Tab. 1 and most of the related work section. We would appreciate more specific feedback and would be happy to comply.
>
> The remark “the idea of using a `classification’ model's heatmap for localization is not particularly new so I would appreciate a detailed discussion” is correct. We have added such a discussion in the revised version.
>
> The revised version describes BLIP in more detail, and we have added a reference to MDETR.

---

> > ### Comment · Reviewer_4wQX · 2022-08-08
> > **Thank you for the rebuttal**
> >
> > Dear Authors,
> >
> > Thank you for your detailed response. My two remaining concern would be:
> >
> > 1) WSOL / WSG is a well studied task with established methods. Thus, it would be better if the author could make it clear how g compares to prior weakly supervised localization method (at a high level; in one or two sentences).
> >
> > 2) My view on the novelty of the task WWbL stays the same: once one has a caption model + a good grounding model, one could coin them together into a WWbL model. The definition of WWbL does not seem to contain any restriction on how we train the model (i.e., whether we can use localization annotations). Thus, we could directly combine many previous models for this purpose. E.g., COCO / VG trained image caption models + MDETR/GLIP. Should such models be compared to the proposed method? Or does WWbL entail no explicit localization annotations?
> >
> > In addition, the evaluation of WWbL is debatble. What the kind of caption we want to model to generate? The current evaluation (in Table 7) evaluates model's ability to generate captions with the same focus as the pointing game datasets. But if the model chooses to generate diverse and creative captions (e.g., if it focuses on novel objects defined in LVIS), should it be penalized? It is also hard to define the granularity of the caption. E.g., when there is a person in the picture, the model can be concise or very detailed (describing each body part separately). A complete evaluation setup should also evaluate the caption quality / diversity / coverage.
> >
> > But I do applaud the fact that the proposed method runs without any localization annotations and rely only on bootstrapping from a classification model's heat map.

---

> > > ### Author Response · Authors · 2022-08-08
> > > **Thank you for the response and the rating update.**
> > >
> > > According to the concerns:
> > >
> > > (1) In comparison to other WSOL/WSG methods, we present a new architecture and loss terms. For example, the loss terms $L_{fore}$ and $L_{back}$ are novel.
> > >
> > > (2) The WWbL is defined in our work as a weakly supervised method. There are two main reasons for doing so: (i) since it is inspired by Marr’s work, we aim to train in an ecological way (the term ecological in this context means in a natural way, as children do), and (ii) we want to have a detection setting that is as general as possible, and specifically, that requires strictly less supervision than WSG.
> > >
> > > (3) We completely agree that the evaluation is not complete. It is based on an established protocol that is retooled from WSG to WWbL. However, this metric neglects some aspects of the problem, which are also neglected in our definition of WWbL. Such neglects are discussed in the context of multiple instances of similar objects (Appendix G of the revised manuscript).

---

### Author Response · Authors · 2022-08-02
**Summary of changes in the revised version.**

We would like to thank all the reviewers for their valuable feedback. We are also grateful for pointing out the typos and the additional experiments to consider, which have been incorporated into the revised manuscript.

[1] Following Reviewer aWBt, we add an experiment that tests the influence of the coefficient controlling $L_{rmap}$ (Appendix C).

[2] Following Reviewer MVej, we add an experiment that tests the influence of the exact CLIP model used to train $g$, on the localization performance of g (appendix H).

[3] Following Reviewer aWBt, we add quantitative results for the scenario where multiple instances of the same object appear in the image (appendix G).

[4] Following Reviewer 4wQX, an additional background for BLIP on related work has been added - line 107 in the revised version.

[5] Following Reviewer 4wQX, MDETR was also added to the related work - line 87

[6] Following Reviewer J9Sc, typos have been corrected.

[7] Following Reviewer aWBt, we add a discussion about biases and ethical issues in the limitation.

[8] Following Reviewers 4wQX and J9Sc, we add our contributions to the introduction.

[9] Following Reviewer MVej, we add the three mentioned papers to the related work.

[10] Following Reviewer 4wQX, we add additional papers that employ GradCam for obtaining pseudo-labels.

[11] Following Reviewer MVej, we have extended the experiment of item [2] above for the phrase grounding task and updated the results in appendix H.

---

### Author Response · Authors · 2022-08-08
**We would be happy to address any follow-up questions**

We thank the reviewers for the detailed feedback and useful ideas. As detailed in the posted summary of changes, we have made an effort to factually address the raised concerns.

We would appreciate the opportunity to discuss our work further if the response to each reviewer has not already addressed all concerns.

---

### Meta-Review · Area_Chair_aArC · 2022-08-27

**Recommendation:** Accept
**Confidence:** Less certain

**Metareview:**

The paper presents a new approach, using two pre-trained models (CLIP and BLIP) as supervision to enable three tasks, including the newly proposed task WWbL, which is a joint open vocabulary description and grounding/localization task trained only with weak supervision.

I recommend acceptance based on the revised paper, the reviewer's comments, and the author response. I think the paper sufficiently contributes:
- Overall idea and architecture
- The WWbL task, even if similar to previous task
- Extensive experimental evaluation and comparison to prior work
- Solid ablation study

The paper received mixed review scores with 2 Borderline rejects, 1 Borderline accept, and 1 strong accept.
The authors have in my opinion largely addressed the concerns and revised the paper, one of the remaining concerns of the weak reject reviewers is novelty, which I think is sufficient.

My recommendation for acceptance is under expectation that the authors revise the paper to address any outstanding points made by reviewers, e.g.
- additional alternative models (reviewer MVej) if possible

Additionally, I think it would be great if the authors discuss the relation ship of WWbL to to the task of dense captioning task more clearly in the paper.

**Award:**

No

---

### Decision · Program_Chairs · 2022-09-14

Accept